# Soluble immune mediators orchestrate protective *in vitro* granulomatous responses across *Mycobacterium tuberculosis* complex lineages

**Ainhoa Arbués[1,2†], Sarah Schmidiger[1,2], Miriam Reinhard[1,2], Sonia Borrell[1,2], Sebastien Gagneux[1,2], Damien Portevin[1,2]***

[1]Swiss Tropical and Public Health Institute, Allschwil, Switzerland; [2]University of Basel, Basel, Switzerland

## eLife Assessment

This study describes the impact of mycobacterial genetic diversity on host-infection phenotypes by assessing the effect of different *M. tuberculosis* lineages on granulomatous inflammation using a 3D in vitro granuloma model. Despite being descriptive and showing mostly correlative relationships, the findings are **useful** and provide some **solid** support regarding the functional impact of M. tuberculosis's natural diversity on host-pathogen interactions. The study will interest researchers working on mycobacteria and motivate future studies to understand how genetic diversity influences virulence and immunity outcomes.

**Abstract** The members of the *Mycobacterium tuberculosis* complex (MTBC) causing human tuberculosis comprise 10 phylogenetic lineages that differ in their geographical distribution. The human consequences of this phylogenetic diversity remain poorly understood. Here, we assessed the phenotypic properties at the host-pathogen interface of 14 clinical strains representing five major MTBC lineages. Using a human *in vitro* granuloma model combined with bacterial load assessment, microscopy, flow cytometry, and multiplexed-bead arrays, we observed considerable intra-lineage diversity. Yet, modern lineages were overall associated with increased growth rate and more pronounced granulomatous responses. MTBC lineages exhibited distinct propensities to accumulate triglyceride lipid droplets—a phenotype associated with dormancy—that was particularly pronounced in lineage 2 and reduced in lineage 3 strains. The most favorable granuloma responses were associated with strong CD4 and CD8 T cell activation as well as inflammatory responses mediated by CXCL9, granzyme B, and TNF. Both of which showed consistent negative correlation with bacterial proliferation across genetically distant MTBC strains of different lineages. Taken together, our data indicate that different virulence strategies and protective immune traits associate with MTBC genetic diversity at lineage and strain level.

## Introduction

With an estimated 1.3 million deaths in 2022, tuberculosis (TB) remains one of the deadliest infectious diseases in the world (***WHO, 2023***). The main etiological agents of TB are a group of closely related bacteria collectively known as the *Mycobacterium tuberculosis* complex (MTBC). Human-adapted MTBC strains are classified into 10 phylogenetic lineages (L1 to L10), which are considered 'ancestral' (L1 and L5 to L10) or 'modern' (L2 to L4) based on the presence or absence of the TbD1 genomic

**\*For correspondence:**
damien.portevin@swisstph.ch

**Present address:** [†]University of Zaragoza, Zaragoza, Spain

**Competing interest:** The authors declare that no competing interests exist.

region (*Brosch et al., 2002*; *Comas et al., 2013*; *Coscolla et al., 2021*; *Guyeux et al., 2024*). Among these lineages, L1 to L4 are responsible for most of the TB cases worldwide, with less prevalent lineages restricted to specific African regions. Local adaptation to specific human hosts has been proposed to support this uneven phylogeographic distribution and disease burden of different MTBC lineages (*Gagneux et al., 2006*). Irrespective of the underlying reason for this phylogeographic population structure, the clinical implications of MTBC genetic diversity, particularly on vaccine efficacy, remain unclear (*Pérez et al., 2020*).

Clinical and epidemiological studies have suggested that MTBC strains might contribute differently to disease presentation and transmissibility, with the difference between ancestral and modern lineages being particularly marked. For example, several reports have described an association between ancestral L1 and higher rates of extrapulmonary TB (*Caws et al., 2008*; *Click et al., 2012*; *Saelens et al., 2022*; *Séraphin et al., 2017*). Moreover, strains belonging to ancestral lineages display a lower capacity to transmit compared to strains from the modern lineages L2 and L4 (*Caws et al., 2008*; *Guerra-Assunção et al., 2015*; *Nebenzahl-Guimaraes et al., 2015*). However, these measures of virulence are influenced by nonbacterial factors, such as host genetics and immune competency or delays in health care seeking or access. The first experimental clues regarding the phenotypic consequences of strain genetic variation in the MTBC date from the 1960s with the observation that South Indian isolates were less virulent than strains from British patients in the guinea pig model (*Mitchison et al., 1960*). Since then, much of our knowledge on the implications of MTBC diversity came from studies characterizing outbreak-causing strains (*Manca et al., 1999*; *Manca et al., 2001*; *Newton et al., 2006*) or focusing on only a few representative strains of specific lineages (*Hiza et al., 2023*; *Romagnoli et al., 2018*). However, extrapolation of strain-specific characteristics to an entire lineage can be misleading due to considerable intra-lineage heterogeneity in experimental outcomes (*Portevin et al., 2011*; *Reiling et al., 2013*; *Wang et al., 2010*). Furthermore, even though awareness of strain variation and its consequences is raising, the majority of immunological studies are based on laboratory-adapted reference strains such as H37Rv or Erdman. In addition, our understanding of immune protective traits in TB remains insufficient to identify good correlates of protection for the assessment of vaccines (*Wang et al., 2024*). In addition, the TB immunology field tends to be dominated by data from mouse and nonhuman primate (NHPs) studies, yet other *in vivo* and *in vitro* models

**Table 1.** *M. tuberculosis* complex (MTBC) isolates used in this work.

| Strain* | | Sublineage† | Patient origin | Additional information |
|---|---|---|---|---|
| L1A | N0069 | L1.1.1 | China | |
| L1B | N0072 | L1.1.2 | India | |
| L1C | N0157 | L.1.2.1 | Philippines | |
| L2A | N0031 | L2.1 | China | Proto-Beijing (DRD105, RD207 present) |
| L2B | N0052 | L.2.2.2 | China | Ancestral Beijing (DRD105, DRD207) |
| L2C | N0155 | L2.2.1 | China | Modern Beijing (DRD105, DRD207, DRD181) |
| L3A | N0004 | L3.1 | India | |
| L3B | N0054 | ND | Ethiopia | |
| L3C | N1274 | L3.2 | Afghanistan | |
| L4A | N1216 | L4.6.2.2 | Ghana | Cameroon—'Specialist' |
| L4B | N0136 | L4.3.3 | USA | LAM—'Generalist' |
| L4C | N1283 | L4.2.1 | Germany | Ural |
| L5A | N1268 | L5.1 | Sierra Leone | |
| L5B | N1272 | L5.1.2 | Ghana | |

*Nomenclature used in the current study (left) and in *Borrell et al., 2019* (right).

†Classification based on *Ates et al., 2018*; *Coll et al., 2014*; *Napier et al., 2020*. ND, not determined; Δ, deletion; RD, region of difference.

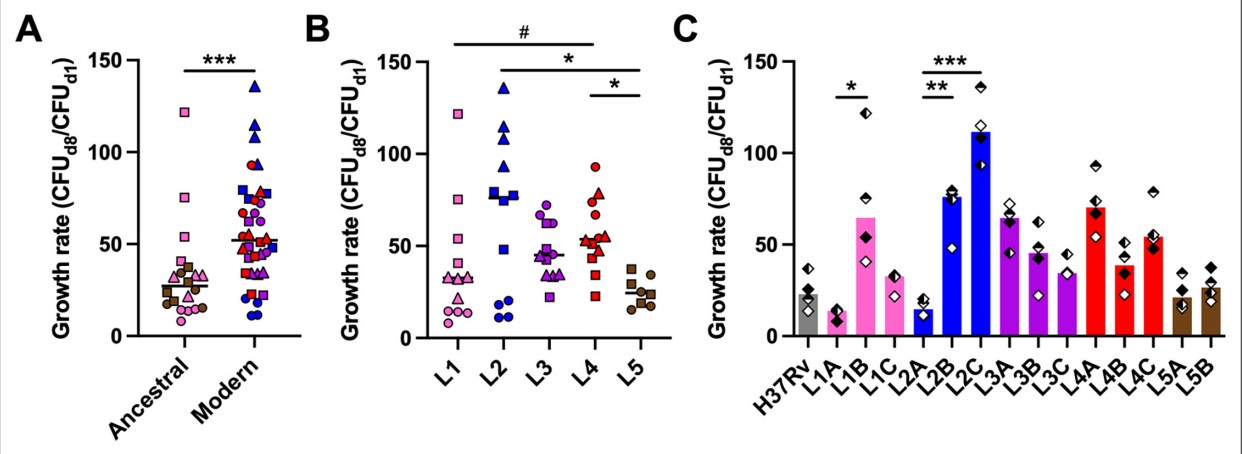

**Figure 1.** Mycobacterial growth in human *in vitro* granulomas shows marked intra-lineage diversity and is overall increased for *M. tuberculosis* complex (MTBC) modern lineages. Growth rate between days 1 and 8 post-infection stratified by (**A**) ancestral or modern lineage, (**B**) specific lineage, or (**C**) individual strain. Colors indicate lineage and horizontal lines and bars represent medians. Shapes stand for (**A, B**) individual strains within the same lineage (LXA in circles, LXB in squares, and LXC in triangles), or (**C**) independent donors (n=4). Statistical analyses by (**A**) two-tailed Mann-Whitney, (**B**) Kruskal-Wallis, or (**C**) Friedman tests with post hoc Dunn's correction. #, $p<0.1$; *, $p<0.05$; **, $p<0.01$; ***, $p<0.001$.

The online version of this article includes the following figure supplement(s) for figure 1:

**Figure supplement 1.** Bacterial load of the selected *M. tuberculosis* complex (MTBC) isolates in human *in vitro* granulomas.

have provided numerous valuable insights (*Esaulova et al., 2021*; *Gideon et al., 2022*; *Moreira-Teixeira et al., 2020*; *Plumlee et al., 2021*; *Tezera et al., 2020*; *Thacker et al., 2020*).

We previously demonstrated the clinical relevance of a three-dimensional (3D) *in vitro* granuloma model by mechanistically dissecting the differential resuscitation of MTB in granulomas exposed to distinct TNF antagonists (*Arbués et al., 2020a*). In this study, we aimed to extend this *in vitro* granuloma model to the study of a large collection of well-characterized and genetically diverse representatives of the MTBC (*Borrell et al., 2019*). We assessed bacterial virulence in terms of growth rate and characterized the individual immune responses to the various MTBC strains. Our results reveal a broad spectrum of granulomatous responses that correlate with the level of mycobacterial proliferation. Overall, strains of the modern lineages are associated with higher growth rates that correlate with increased macrophage cell death and aggregate formation scores (based on size and shape); nevertheless, we also report substantial intra-lineage heterogeneity. Furthermore, our results show that granulomatous responses associated with the least replicating strains harbor an increased CD4 and CD8 T cell activation as well as the release of specific soluble factors encompassing CXCL9, granzyme B, and TNF.

## Results

### Mycobacterial growth in human *in vitro* granulomas shows marked intra-lineage diversity and is overall increased for MTBC modern lineages

We selected 14 isolates (*Table 1*) from a reference set of clinical strains covering much of the global diversity of the human-adapted MTBC (*Borrell et al., 2019*). As a first measure of virulence, we evaluated the capacity of the strains to proliferate within 3D *in vitro* granulomas. Bacterial load based on colony forming units (CFU) was quantified on days 1 and 8 post-infection (*Figure 1—figure supplement 1*) and used to determine the growth rate between these time points. The different strains proliferated with rates ranging from 8.1 to 136.0 (*Figure 1*). Modern lineages (L2 to L4) proliferated significantly more than ancestral ones (L1 and L5) did (median rates 52.1 [modern] vs. 27.2 [ancestral]) (*Figure 1A*). Stratification of the data by lineage revealed that the growth rate of L2 tended to be higher than that of the other modern lineages (median rates 76.0 [L2] vs. 45.1 [L3] and 53.7 [L4]) (*Figure 1B*). Nevertheless, substantial intra-lineage heterogeneity could be observed, particularly within L1 and

L2 (coefficients of variation 84.4% [L1] and 66.0% [L2] vs. 32.6% [L3], 34.6% [L4], and 31.9% [L5]). As depicted in *Figure 1C*, the lower virulence observed for ancestral lineages in general was contrasted by strain L1B that displayed a high growth rate (median 64.7) comparable to that of most strains belonging to modern lineages. Moreover, while L2 Beijing strains (L2B and L2C) exhibited a characteristic hyper-virulent phenotype (median rates 76.0 and 111.6, respectively), the proto-Beijing L2A was one of the most attenuated strains (median 14.7). In comparison, L3 to L5 representative strains behaved more homogeneously. Noteworthy, the reference strain H37Rv, belonging to L4, displayed a lower growth rate than any of the evaluated L4 clinical isolates. Taken together, our results confirm the higher virulence propensities of strains from modern lineages, and L2 Beijing strains in particular, although they also highlight considerable intra-lineage diversity.

## Propensity to enter a dormant-like state is particularly pronounced in lineage 2 and reduced in lineage 3 strains

MTBC pathogenicity relies on its ability to survive the hostile and hypoxic granulomatous environment. The transcription factor DosR orchestrates the adaptation to oxygen limitation by inducing a metabolic switch into a dormant state (*Garton et al., 2008*; *Gengenbacher and Kaufmann, 2012*). A constitutive overexpression of *dosR* in L2 Beijing strains has been proposed to contribute to their enhanced virulence and transmissibility (*Reed et al., 2007*). A genomic duplication including the DosR operon and a synonymous SNP within Rv3134c, which generates an alternative transcriptional start site, have been linked to this increased expression of the DosR regulon (*Domenech et al., 2010*; *Domenech et al., 2017*). A hallmark of our *in vitro* granuloma model resides in the generation of a hypoxic environment leading to dormant-like MTBC features such as the induction of DosR-regulated genes and the loss of acid-fastness and accumulation of triacylglycerides, which can be quantified by differential Auramine-O (Au)/Nile red (NR) fluorescent staining (*Arbués et al., 2020a*; *Arbués et al., 2021*; *Kapoor et al., 2013*). We assessed whether MTBC genetic diversity would result in a differential propensity to enter dormancy when facing a hypoxic environment. Therefore, we quantified the ratio between dormant-like (Au$^-$ NR$^+$) and metabolically active (Au$^+$ NR$^-$) bacteria (hereafter referred to as 'dormancy ratio') for each MTBC strain (*Figure 2A*, *Figure 2—figure supplement 1A, B*). While approximately 50% of H37Rv bacilli acquired dormant-like features (median dormancy ratio 0.94), MTBC clinical strains showed a level of dormancy induction spanning from 0.29 to 1.98. Upon grouping strains by lineage (*Figure 2B*, *Figure 2—figure supplement 1C*), no significant difference was found between L1, L4, and L5 (median dormancy ratios 0.85 [L1], 0.86 [L4], and 1.06 [L5]). Nonetheless, L2 strains exhibited an increased propensity to accumulate triacylglycerides (median dormancy ratio 1.30; p=0.072 vs. L1 and p<0.05 vs. L4 by corrected Dunn's multiple comparisons test). This result is in line with the fact that they are carriers of the aforementioned genomic events: the duplication in L2A, the Rv3134c SNP in L2B, and both in L2C. By contrast, L3 bacilli seemed more prone to remain in a metabolically active state in comparison to bacteria belonging to the other lineages (median dormancy ratio 0.54; p=0.072 vs. L5 by corrected Dunn's multiple comparisons test). Interestingly, the ratio between dormant-like and metabolically active bacilli showed no correlation with the respective growth rate of the individual MTBC strains (*Figure 2C*). This finding suggests that an increased tendency to remain in a metabolically active state is not necessarily associated with a more pronounced growth phenotype. Further studies are necessary to better understand the dormancy ratio-growth rate relationship, and to assess how this might relate to *in vivo* virulence.

## MTBC strain diversity translates into a spectrum of granulomatous responses that correlates with bacterial growth and induction of macrophage apoptosis

We next investigated the impact of MTBC strain diversity on the host immune responses within *in vitro* granulomas. We monitored cellular aggregation by bright-field microscopy imaging on day 7 post-infection and quantified the area and aspect ratio of the individual aggregates as well as their total number. While uninfected (UI) peripheral blood mononuclear cells (PBMCs) did not lead to significant aggregation, all MTBC strains induced the formation of *in vitro* granulomas in variable numbers, shapes, and sizes (*Figure 3A*). Despite notable donor-to-donor variability, the differential granulomatous responses induced by the individual strains were consistent across all donors (*Figure 3—figure supplement 1*). Overall, 10–58 *in vitro* granulomas were identified per field (*Figure 3—figure*

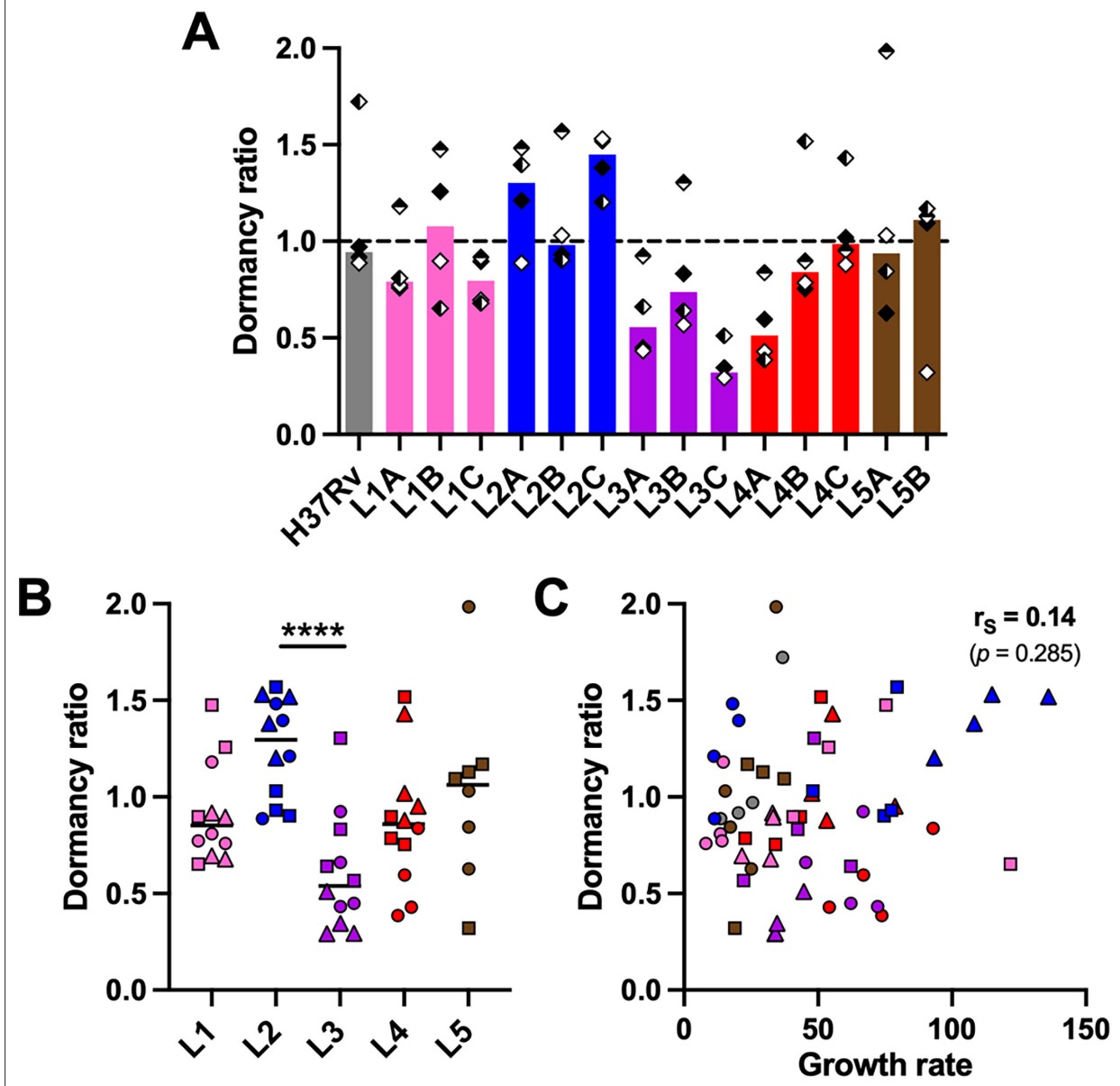

**Figure 2.** Propensity to enter dormancy is particularly pronounced in lineage 2 and reduced in lineage 3 strains. Bacilli recovered on day 8 post-infection were stained with Auramine-O (Au) and Nile red (NR) and quantified by fluorescence microscopy. Dormancy ratio was defined as the ratio between dormant-like (Au⁻ NR⁺) and metabolically active (Au⁺ NR⁻) bacteria. Data stratified by (**A**) individual strain or (**B**) lineage. (**C**) Two-tailed Spearman's correlation analysis of dormancy ratio with growth rate. Colors indicate lineage and bars and horizontal lines represent medians. Shapes stand for (**A**) independent donors (n=4) or (**B, C**) individual strains within the same lineage (LXA in circles, LXB in squares, and LXC in triangles). Statistical analyses by (**A**) Friedman or (**B**) Kruskal-Wallis tests with post hoc Dunn's correction. ****, p<0.0001.

The online version of this article includes the following figure supplement(s) for figure 2:

**Figure supplement 1.** Auramine-O (Au)/Nile red (NR) profile of the selected *M. tuberculosis* complex (MTBC) isolates.

*supplement 2A*). Strains L1A, L2A, and L5B displayed a tendency to induce more aggregates per field (median 42.5 [L1A], 40 [L2A], and 31.5 [L5B] vs. 22 [across all strains]) that were characterized by a less circular shape (i.e. higher aspect ratio) (median aspect ratios 1.38 [L1A], 1.35 [L2A], and 1.33 [L5B] vs. 1.25 [across all strains]) (*Figure 3—figure supplement 2B*). By contrast, L3 and L4 strains triggered the formation of markedly larger granulomas (median average area per granuloma in μm²: 6586.4 [L3] and 7800 [L4] vs. 3377.8 [L1], 5199.2 [L2] and 4638.4 [L5]) (*Figure 3—figure supplement 2C*). In order to simplify the ensuing analyses, we integrated these parameters into an aggregate formation score

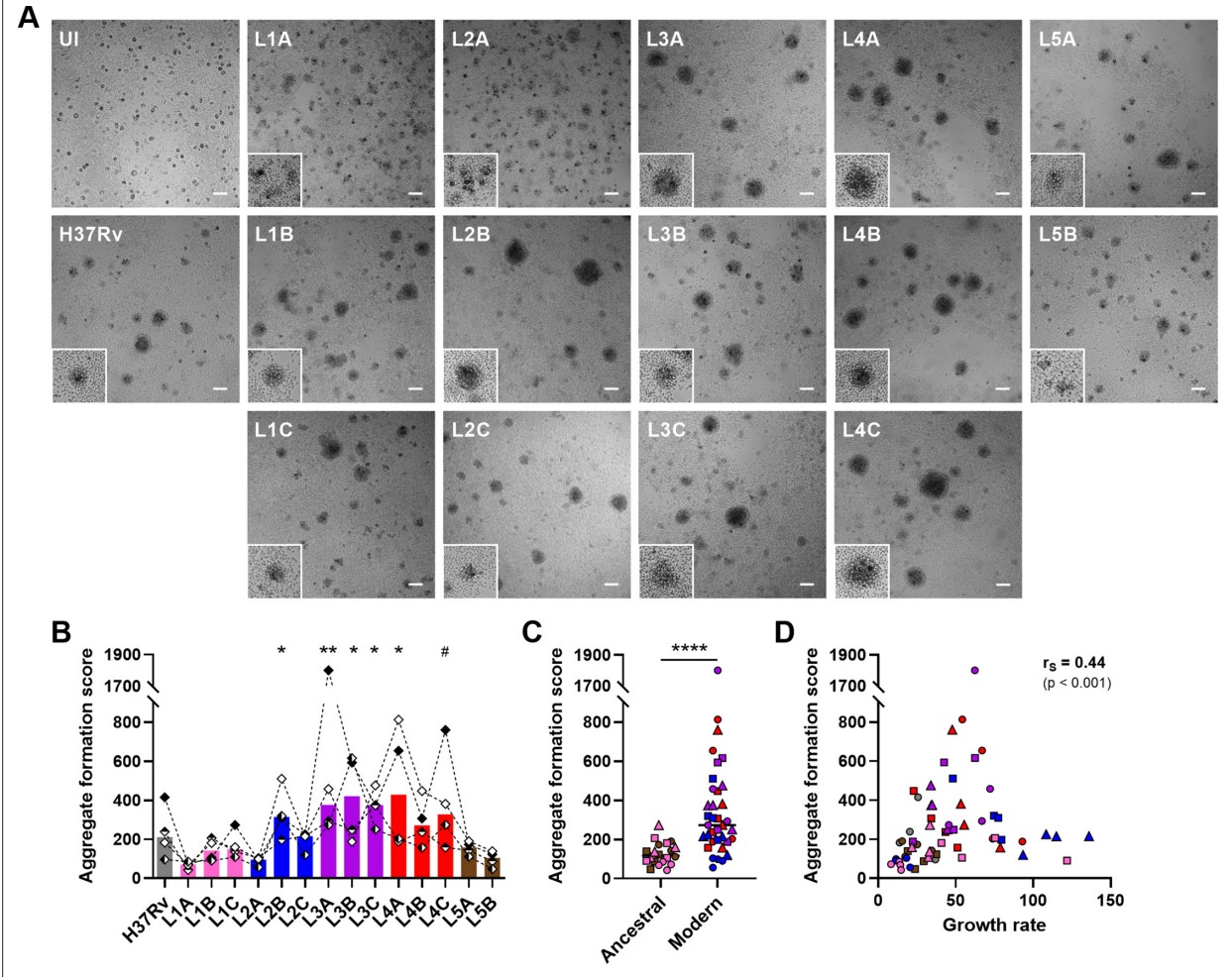

**Figure 3.** *M. tuberculosis* complex (MTBC) strain diversity translates into a spectrum of granulomatous responses that correlates with bacterial growth. (**A**) Bright-field images of *in vitro* granulomas on day 7 post-infection from a representative donor. Scale bar = 100 µm. UI, uninfected. Lower left corner higher magnification inserts (200 µm sides) depict granulomas of characteristic size and morphology. (**B–D**) Number, area, and aspect ratio of cell aggregates were quantified and integrated into an aggregate formation score (see Materials and methods section). Aggregate formation scores stratified by (**B**) individual strain and (**C**) ancestral or modern lineage. (**D**) Two-tailed Spearman's correlation analysis of aggregate formation score with growth rate. Colors indicate lineage and bars represent medians. Shapes stand for (**B**) independent donors (n=4) or (**C, D**) individual strains within the same lineage (LXA in circles, LXB in squares, and LXC in triangles). Statistical analyses by (**B**) Friedman test with post hoc Dunn's correction (comparisons against L1A) or (**C**) two-tailed Mann-Whitney test. #, $p < 0.1$; *, $p < 0.05$; **, $p < 0.01$; ****, $p < 0.0001$.

The online version of this article includes the following figure supplement(s) for figure 3:

**Figure supplement 1.** Granulomatous response variability across *M. tuberculosis* complex (MTBC) strains is consistent across independent donors.

**Figure supplement 2.** The number, shape, and size of *in vitro* granulomas vary depending on the *M. tuberculosis* complex (MTBC) infecting strain.

(see Materials and methods section) that reflects the gamut of responses observed so that the bigger and more circular the granulomas, the higher the aggregate formation score (*Figure 3B*). Aggregate formation scores ranged from 42.2 to 1801.1, with a median of 199.5. Infection with modern lineages, especially L3 and L4, led to granulomatous responses with significantly higher scores (median scores 205.6 [L2], 374.8 [L3], and 290.2 [L4] vs. 107.2 [L1] and 130.2 [L5]) (*Figure 3C*). Interestingly, the aggregate formation score showed a significant positive correlation with the respective strain growth rate (*Figure 3D*).

It has been reported that apoptosis of infected macrophages drives early dissemination and granuloma formation (*Aguilo et al., 2013*; *Davis and Ramakrishnan, 2009*). Therefore, we sought to evaluate macrophage apoptosis induced by the various strains in the context of granulomatous responses using flow cytometry (*Figure 4—figure supplement 1*). We did not observe significant differences

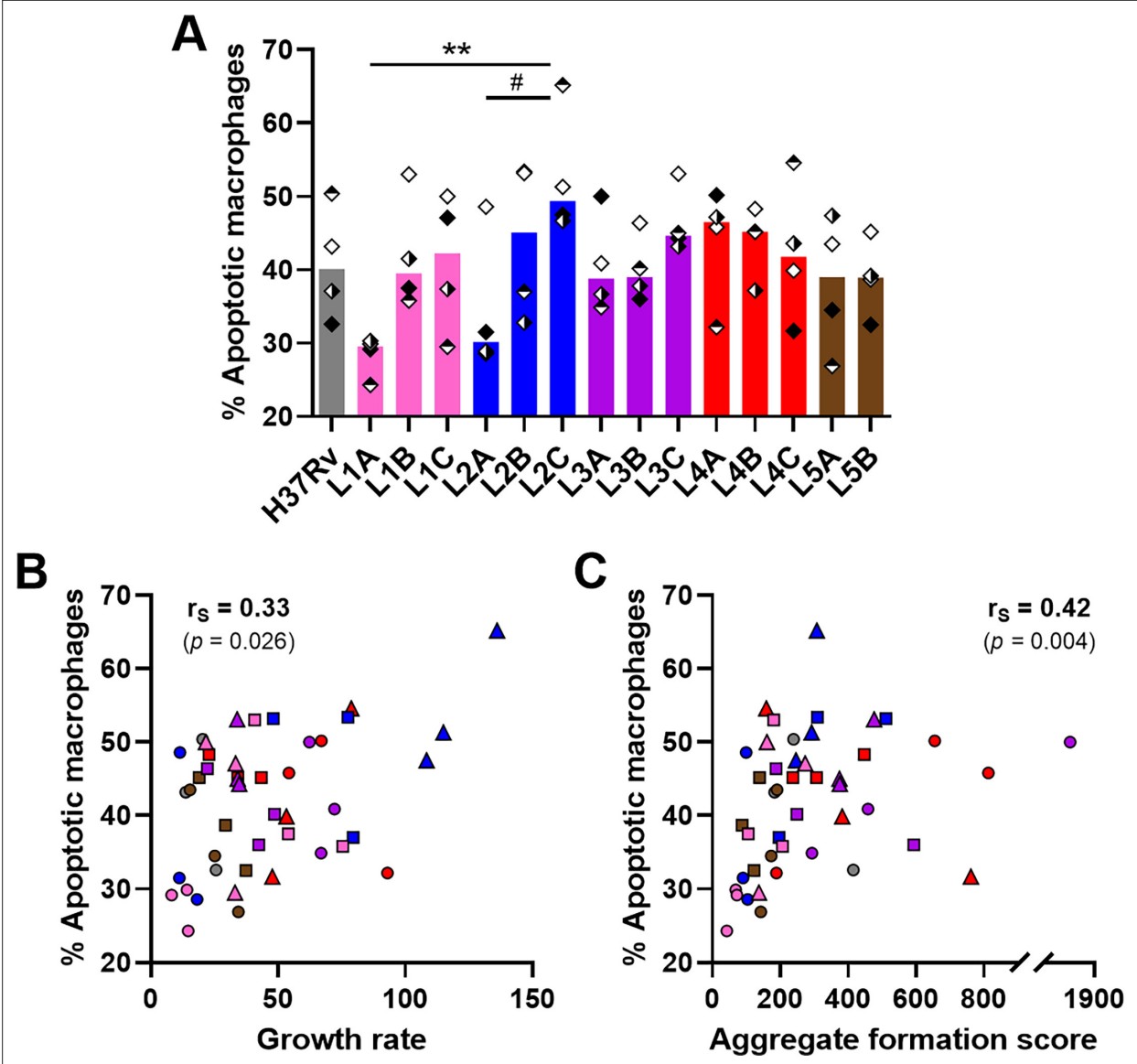

**Figure 4.** Macrophage apoptosis is positively associated with mycobacterial growth rate and aggregate formation score. (**A**) Percentage of apoptotic macrophages (CD11b⁺ Annexin V⁺ 7-AAD⁻) on day 6 post-infection quantified by flow cytometry. (**B, C**) Two-tailed Spearman's correlation analysis of macrophage apoptosis with (**B**) growth rate or (**C**) aggregate formation score. Colors indicate lineage and bars represent medians. Shapes stand for (**A**) independent donors (n=4) or (**B, C**) individual strains within the same lineage (LXA in circles, LXB in squares, and LXC in triangles). (**A**) Statistical analysis by Friedman test with post hoc Dunn's correction. #, p<0.1; **, p<0.01.

The online version of this article includes the following figure supplement(s) for figure 4:

**Figure supplement 1.** Flow cytometry gating strategy for the quantification of cell death induction in CD11b⁺ macrophages.

**Figure supplement 2.** *M. tuberculosis* complex (MTBC) strains did not induce significantly different percentages of necrotic macrophages.

in the total number of CD11b⁺ macrophages recovered 6 days after infection with the various MTBC strains nor in the percentage of necrotic macrophages (Annexin V⁺ 7-AAD⁺) (*Figure 4—figure supplement 2*). Overall, the percentage of apoptotic macrophages (Annexin V⁺ 7-AAD⁻) ranged from 24.3% to 65.2%, with a median of 41.2% (*Figure 4A*). Most of the strains shared intermediate levels of macrophage apoptosis, including the reference strain H37Rv (median percentage of apoptotic macrophages 40.2%). By contrast, L1A and L2A strains, which were associated with the lowest growth rates and granulomatous responses, also exhibited a reduced induction of macrophage apoptosis (median percentages 29.6% and 30.2%, respectively); while L2C, the most virulent strain, induced the highest apoptosis levels (median percentage of apoptotic macrophages 49.4%). In fact, the percentage of

apoptotic macrophages showed a significant positive correlation with the replication rate of the respective infecting MTBC strain (*Figure 4B*) as well as with aggregate formation scores (*Figure 4C*). Taken together, our results suggest that strains exhibiting greater proliferation are more prone to induce macrophage apoptosis and a stronger granulomatous response.

## Rapid lymphocyte activation as well as CXCL9, granzyme B, and TNF secretion are associated with reduced mycobacterial growth across the MTBC

We next sought to profile the immune granulomatous responses induced by the MTBC strains and relate them to their respective growth rate. We first monitored by flow cytometry lymphocyte proliferation using carboxyfluorescein succinimidyl ester (CFSE) dilution assay, as well as lymphocyte activation based on the expression of surface markers CD69, CD25, HLA-DR, and CD38 on day 6 post-infection (*Figure 5—figure supplement 1*). We focused our analysis on CD69-, CD38-, CD25-, and HLA-DR-expressing CD4 T cells, along with CD69- and CD38-expressing CD8 T cells, whose frequency increased consistently across all donors upon infection with at least one strain (compared to the uninfected control) (*Figure 5—figure supplement 2A, B*). Markedly stronger T cell responses were detected for one of the donors. This observation was consistent with a higher frequency of mycobacteria-specific CD4 T cells recalled by PPD stimulation (*Figure 5—figure supplement 2C*). To account for inter-donor variability, we scaled the responses of every donor dividing the individual response to each strain by the average response across all strains of that same donor (*Figure 5A*). As depicted in *Figure 5B*, granulomatous responses elicited by L1A, L2A, and both L5 strains entailed marked CD4 T cell proliferation. Across all MTBC strains, proliferation was consistently associated with an increased expression of the assessed activation markers on both CD4 and CD8 T cells. Nevertheless, most activated CD4 T cells expressed only one of the studied markers (*Figure 5C*). Overall, ancestral lineages were the most potent inducers of all the aforementioned T cell populations (*Figure 5D*, *Figure 5—figure supplement 3A*), with CD38-expressing CD4 T cells constituting the most differentially induced population (median scaled frequencies 1.21 [ancestral] vs. 0.79 [modern]). Among modern lineages, L4 induced the strongest T cell activation (*Figure 5E*, *Figure 5—figure supplement 3B*), which actually was not statistically different from that elicited by ancestral strains (*Figure 5D*, *Figure 5—figure supplement 3A*). Infection with L4 strains resulted in significantly higher percentages of CD38-positive CD4 T cells compared to L2 Beijing strains (median scaled frequencies 0.94 vs. 0.54, respectively), while L3 strains elicited intermediate responses (median scaled frequency 0.73). When considering bacterial growth, strains eliciting an enhanced T cell response appeared among the ones replicating the least. We computed Spearman's rank correlation tests with Benjamin-Hochberg multiple testing correction between CD4 and CD8 T cell responses and the growth rate (*Figure 5F*). CD4 T cell proliferation as well as all expression of all activation markers exhibited significant, moderate to strong, negative correlations with MTBC strain growth rate, with the expression of CD38 resulting in the strongest coefficient. Activation of CD8 T cells, determined by the expression of either CD38 or CD69, was also significantly associated with a hampered mycobacterial proliferation. Shedding light on what distinguishes granulomas that are protective from those that support bacterial growth, a significant negative correlation between macrophage apoptosis induction and T cell activation can be observed specifically with activated CD4 T cells expressing CD38 ($r_S$ = –0.36, p<0.05) or CD69 ($r_S$ = –0.40, p<0.01). Taken together, our data indicate that stronger T cell activation is associated with MTBC isolates exhibiting low bacterial replication, irrespective of the lineage they belong to.

Subsequently, we investigated whether a panel of soluble immune factors—linked or not to T cell activation—may also be associated with a reduced bacterial growth. We focused our analysis on molecules present in granuloma supernatants on day 1 or 8 post-infection that were modulated by the infection (*Figure 6—figure supplement 1*). We again scaled the data to account for inter-donor variability, as described above (*Figure 6A*). Differentially modulated molecules included the innate cytokines TNF and IL-1β at both time points, IL-10 and the chemokine CXCL9 at the earlier time point, and IFN-γ, IL-17 (A and F), IL-22, IL-13, and granzyme B at the later (*Figure 6B*). Among these, IL-1β and IL-13 displayed significant lineage-associated trends. L4 and, to a lesser extent, L3 induced a significantly higher secretion of IL-1β on day 1 post-infection (median scaled responses 1.54 [L4] and 1.31 [L3] vs. 0.55 [L1], 0.94 [L2], and 0.89 [L5]) (*Figure 6C*, *left*). This pattern remained at the

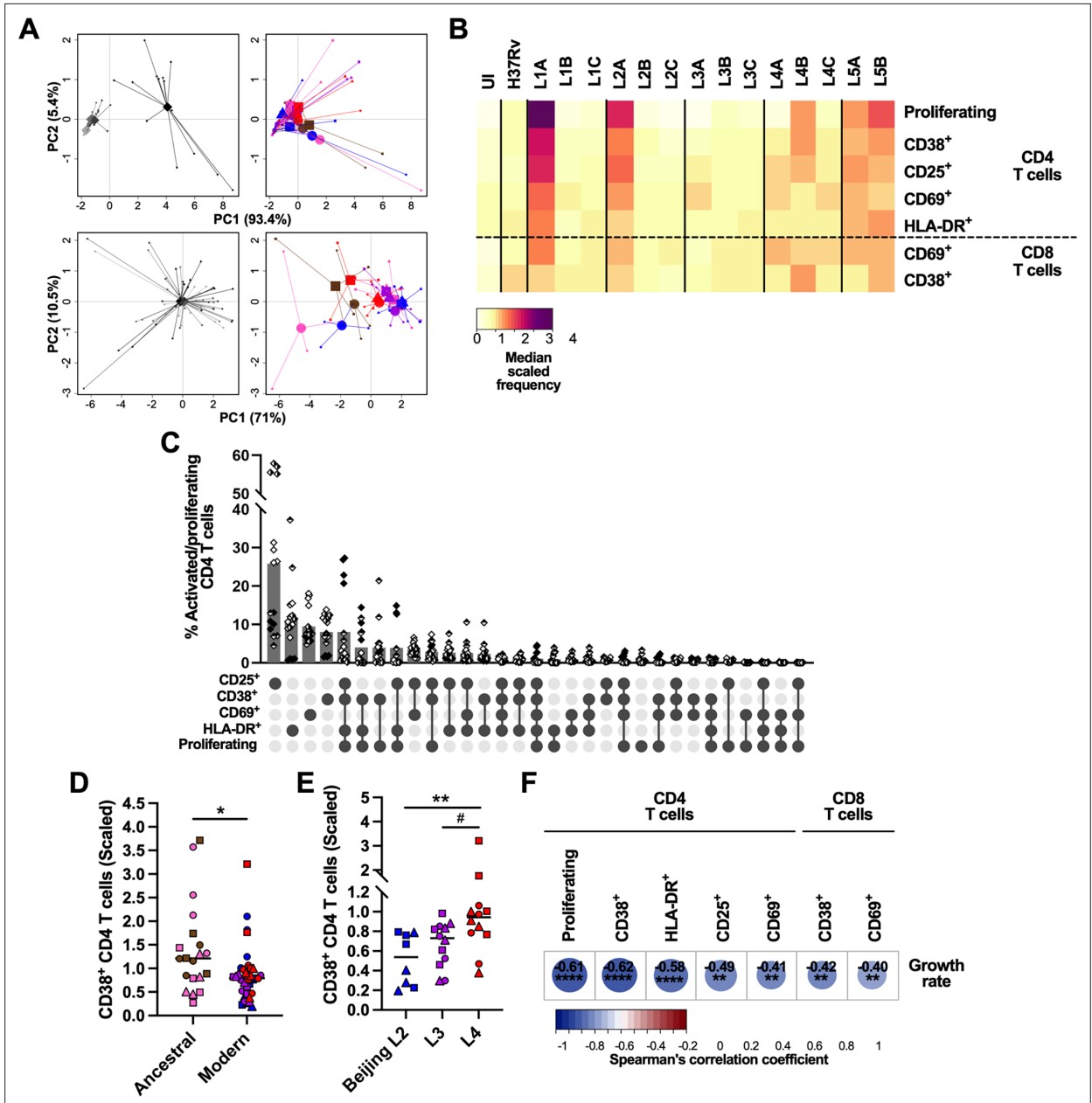

**Figure 5.** Activation and proliferation of T cells is associated with reduced mycobacterial growth across the *M. tuberculosis* complex (MTBC). Percentage of activated (expressing any of the markers) and proliferating (carboxyfluorescein succinimidyl ester [CFSE⁻]) T cells on day 6 post-infection was quantified by flow cytometry. (**A**) Principal component (PC) analysis using raw (top panels) or scaled (lower panels) data. Data were grouped by donor (left panels), with gray shades representing independent donors (n=4); or by MTBC infecting strain (right panels), with colors indicating lineage and shapes standing for individual strains within the same lineage (LXA in circles, LXB in squares, and LXC in triangles). (**B**) Heatmap representing the median scaled frequencies induced by each MTBC strain. UI, uninfected. (**C**) Marker co-expression on the activated/proliferating CD4 T cells induced by L1A, L2A, and L5 strains. Bars represent the mean percentage of cells expressing the indicated marker combination and each filling pattern of the shapes corresponds to an individual donor. (**D, E**) Percentage of CD38⁺ CD4 T cells elicited by (**D**) ancestral or modern lineages, or (**E**) specific modern lineages. Colors and shapes same as in (**A**) and horizontal lines represent medians. Statistical analyses by (**D**) two-tailed Mann-Whitney or (**E**) Kruskal-Wallis tests with post hoc Dunn's correction. (**F**) Heatmap of Spearman's correlation coefficients of activated/proliferating T cell populations with MTBC growth rate (two-tailed, post hoc Benjamini-Hochberg corrected). #p<0.1; *, p<0.05; **, p<0.01; ****, p<0.0001.

The online version of this article includes the following figure supplement(s) for figure 5:

**Figure supplement 1.** Flow cytometry gating strategy for the analysis of T cell proliferation and activation.

**Figure supplement 2.** *M. tuberculosis* complex (MTBC) strain diversity results in induction of quantitatively distinct T cell responses.

**Figure supplement 3.** Induction of T cell activation and proliferation exhibits both lineage- and mycobacterial growth-associated trends.

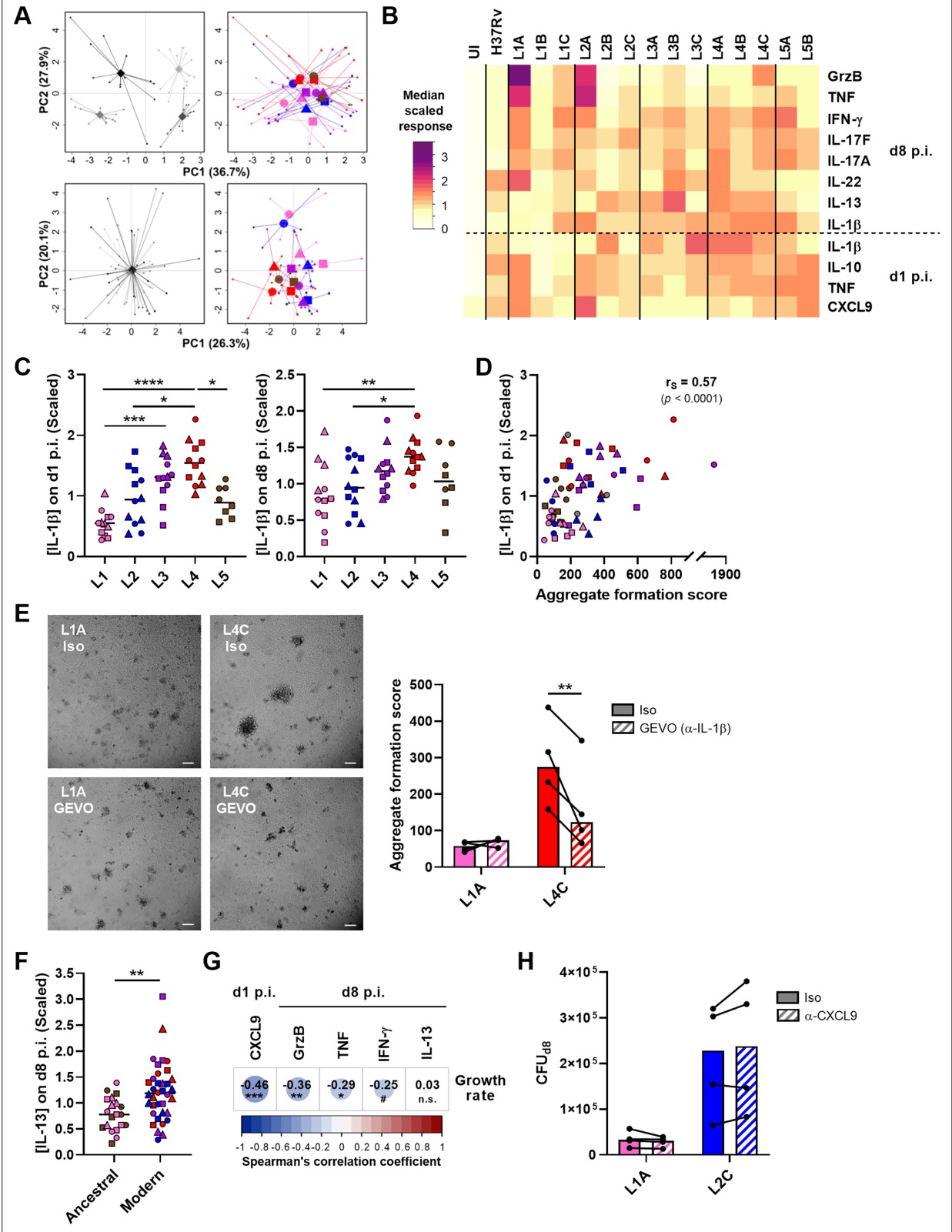

**Figure 6.** CXCL9, granzyme B, and TNF secretion is associated with reduced mycobacterial growth across the *M. tuberculosis* complex (MTBC). The concentration of soluble mediators in the supernatant of *in vitro* granulomas on days 1 and 8 post-infection (p.i.) was quantified by multiplex bead-based immunoassay. (**A**) Principal component (PC) analysis was performed using raw (top panels) or scaled (lower panels) concentrations. Data were grouped by donor (left panels), with gray shades representing independent donors (n=4); or by MTBC infecting strain (right panels), with

*Figure 6 continued on next page*

*Figure 6 continued*

colors indicating lineage and shapes standing for individual strains within the same lineage (LXA in circles, LXB in squares, and LXC in triangles). (**B**) Heatmap of the median scaled response induced by each MTBC strain. UI, uninfected; GrzB, granzyme B. (**C**) IL-1β response stratified by lineage. Colors and shapes same as in (**A**) and horizontal lines represent medians. (**D**) Two-tailed Spearman's correlation analysis of IL-1β response on day 1 p.i. with aggregate formation score. Colors and shapes same as in (**A**). (**E**) Effect of blocking of IL-1β on granuloma formation using gevokizumab (GEVO). Bright-field images of *in vitro* granulomas on day 7 p.i. from a representative donor. Iso, isotype control. Scale bar = 100 µm. Aggregate formation score was calculated as indicated in *Figure 3*. Colors indicate lineage, bars represent medians, and data from the same donor are connected through lines. (**F**) IL-13 response stratified by ancestral or modern lineages. Colors and shapes same as in (**A**) and horizontal lines represent medians. (**G**) Heatmap representing Spearman's correlation coefficients of cytokine concentrations with MTBC replication rate (two-tailed, post hoc Benjamini-Hochberg corrected). (**H**) Effect of early blocking of CXCL9 on bacterial load on day 8 p.i. Colors indicate lineage, bars represent medians, and data from the same donor are connected through lines. Statistical analyses by (**C**) Kruskal-Wallis test with post hoc Dunn's correction, (**E, H**) two-way ANOVA with post hoc Sidak's correction, or (**F**) two-tailed Mann-Whitney tests. n.s., p>0.1; #, p<0.1; *, p<0.05; **, p<0.01; ***, p<0.001; ****, p<0.0001.

The online version of this article includes the following figure supplement(s) for figure 6:

**Figure supplement 1.** *M. tuberculosis* complex (MTBC) diversity results in variable soluble factor immune responses.

later time point for L4 (median scaled responses 1.37 [L4] vs. 0.79 [L1], 0.95 [L2], 1.17 [L3], and 1.03 [L5]) (*Figure 6C*, *right*). Interestingly, the IL-1β response on day 1 post-infection showed a moderate positive correlation with aggregate formation score (*Figure 6D*). In order to investigate the influence of IL-1β levels on the granulomatous response, IL-1β was blocked throughout the infection using gevokizumab (*Issafras et al., 2014*) and granuloma formation was monitored on day 7 post-infection. Neutralization significantly decreased the score of granulomas generated by high IL-1β inducer strain L4C, whereas it had no effect on granulomas associated with low IL-1β inducer strain L1A, thereby establishing a cause-effect relationship between these two phenotypes (*Figure 6E*). Moreover, we found that the CD4 T helper 2 cytokine IL-13 was preferentially induced in the context of infection with modern lineages compared to ancestral lineages (median scaled responses 1.18 [modern] vs. 0.78 [ancestral]) (*Figure 6F*). While we found no significant correlation for IFN-γ, induction of CXCL9 secretion on day 1 post-infection, and granzyme B as well as TNF on day 8 post-infection were associated with a reduced MTBC bacterial proliferation (*Figure 6G*). However, blocking of CXCL9 during the infection failed to confirm causality since it did not impact bacterial load of either the high CXCL9 inducer strain L1A or the low CXCL9 inducer strain L2C (*Figure 6H*). Altogether, these results convey that a robust T cell activation characterized by an early CXCL9 detection and higher levels of TNF and granzyme B may orchestrate the control of mycobacterial growth.

## Discussion

There is increasing epidemiological and experimental evidence that MTBC strain diversity may influence pathogenicity. Yet, only a small number of studies based on human specimens have compared a diverse range of strains belonging to more than two lineages (*Portevin et al., 2011*; *Reiling et al., 2013*). Here, we combine an *in vitro* granuloma model (*Arbués et al., 2020a*; *Arbués et al., 2021*) with a set of representative L1 to L5 strains (*Borrell et al., 2019*) to demonstrate that MTBC genetic variation affects the crosstalk with human immune cells. The model encompasses all PBMC-derived cell types involved in TB immune responses, but lacks granulocytes (i.e. neutrophils, eosinophils, basophils, and mast cells). Consequently, the interface between the MTBC strains and neutrophils—which play a pathogenic role in TB by providing a permissive environment for mycobacterial replication (*Andrews et al., 2024*; *Gideon et al., 2019*; *Lowe et al., 2013*)—could not be investigated. We show that MTBC isolates display a range of growth rates correlating with the induction of macrophage apoptosis and the extent of the resulting granulomatous structures. This result is in agreement with the current view of the tuberculous granuloma as a structure that, beyond functioning as purely host-protective, is exploited by pathogenic mycobacteria as a niche for proliferation and dissemination (*Pagán and Ramakrishnan, 2014*). Our data reveal various features that associate with certain MTBC lineages and/or strains. Noteworthy, we have identified immune response traits that correlate with hampered mycobacterial proliferation.

'Modern' lineages (L2 to L4) are often associated with globally spread TB epidemics, whereas TbD1-intact 'ancestral' lineages (L1 and L5 to L10) rather represent endemic strains restricted to a given geographical area. A general consensus arose that strains belonging to modern lineages would

exhibit enhanced capacities to proliferate compared to those from ancestral ones (*Reiling et al., 2013*; *Romagnoli et al., 2018*). Actually, in an elegant work using recombinant strains, Bottai and colleagues demonstrated that TbD1 deletion significantly increases MTB virulence in relevant infection models (*Bottai et al., 2020*). In line with these previous studies, ancestral lineages display reduced virulence in *in vitro* granulomas compared to the modern lineages. L1 is found in countries along the rim of the Indian Ocean. This geographical confinement together with experimental evidence (*Bottai et al., 2020*; *Reiling et al., 2013*; *Romagnoli et al., 2018*) prompted a general view of L1 as a low virulence lineage. Our data showed that, overall, L1 exhibited lower virulence than modern lineages. Nonetheless, we observed a significant intra-lineage variation, with strain L1C exhibiting a growth rate comparable to those of strains belonging to modern lineages. This phenotypic variability may be a reflection of the highly diverse population structure exhibited by this lineage (*Netikul et al., 2021*). In line with our findings, Mitchison and colleagues also documented a wide range of virulence among South Indian isolates, with one-third of them being as virulent as the British ones (*Mitchison et al., 1960*). Even though it is unclear whether all those strains belonged to L1, we can assume that a majority of them were when considering epidemiological data (*Poonawala et al., 2020*). These results exposing virulence variation across L1 isolates challenge the 'low virulence' attribute generally given to L1. However, a recent report revealed that the contribution of L1 to the global TB pandemic is generally undervalued (*Netikul et al., 2021*). Due to its prevalence in countries with the highest TB incidence, L1 was estimated to account for 28% of the total TB cases. Interestingly, the strains with the higher replication rates, L1B and L1C, were isolated from patients of Indian and Filipino origin and belong to the most common sublineages, L1.1.2 and L1.2.1, respectively. Taken together these data suggest that some L1 sublineages may have evolved mechanisms to compensate for the attenuation associated to the presence of TbD1 (*Bottai et al., 2020*). Meanwhile, L5, previously known as *Mycobacterium africanum* 1, is geographically restricted to West Africa. Experimental data on L5 are scarce and most of our understanding about TB caused by *M. africanum* derives from the study of L6, previously reported as *M. africanum* 2 (*Silva et al., 2022*). In contrast to the downregulation of the DosR regulon reported in the sputum of L6-infected patients compared to L4 (*Ofori-Anyinam et al., 2017*), we observed a similar induction of dormant-like mycobacteria in L4 and L5. In accordance with their ancestral genotype, both L5 strains displayed comparable low replication rates and elicited increased frequencies of activated T cells. Nevertheless, we cannot dismiss a potential contribution of host genetics to the apparent attenuation of L5. In Ghana, infection by L5 has been associated with the Ewe ethnic group (*Asante-Poku et al., 2015*). L5 strains displayed lower growth in macrophages from the Akan ethnic group; still, proliferation in Ewe's macrophages was comparable to that of L4 isolates (*Osei-Wusu et al., 2023*).

The three modern lineages (L2 to L4) displayed comparable growth rates overall. Again, substantial strain-to-strain variability was observed within L2. The L2 Proto-Beijing strain L2A was markedly attenuated to a level comparable to that of the ancestral lineages. At the other end of the virulence spectrum, the representative of the most recently evolved L2 strains, the so-called 'modern' L2 Beijing, displayed the highest proliferative capacity across all the MTBC strains studied. For its part, the 'ancestral' Beijing isolate exhibited a bacterial burden comparable to those of strains belonging to other modern lineages. This finding suggests that L2 strains have evolved toward increased virulence. This is also consistent with prior studies showing that modern Beijing strains exhibited enhanced virulence in mice compared to those with an ancestral genotype (*Ribeiro et al., 2014*). The L2 Beijing sublineage has received particular attention due to its high transmissibility and virulence in cellular and animal models (*Guerra-Assunção et al., 2015*; *Manca et al., 2001*; *Tram et al., 2018*; *Tsenova et al., 2005*). One previously identified mechanism contributing to the hyper-virulent phenotype of L2 Beijing strains is the inhibition of the production of pro-inflammatory cytokines (*Reed et al., 2004*; *Wang et al., 2010*). Likewise, we found that both L2 Beijing strains were poor inducers of T cell activation and pro-inflammatory cytokines. By contrast, we found the high replication rate of L4 strains to be accompanied by a significant T cell activation and high levels of IFN-γ. These data are consistent with the fact that MTBC strain CDC1551, responsible for one of the most studied L4 outbreaks, was characterized by very large skin test responses in infected subjects and a robust pro-inflammatory immune response in mice and in human monocytes (*Manca et al., 1999*). In addition, we observed that L4 and, to a lesser extent, L3 promoted an increased secretion of IL-1β. Romagnoli and collaborators showed that higher IL-1β levels trigger autophagy;

however, L4 isolates evade subsequent lysosomal degradation (*Romagnoli et al., 2018*). Despite its protective role (*Mayer-Barber et al., 2010*), increased IL-1β levels in the bronchoalveolar lavage or serum of TB patients have been associated with presence of cavities and higher bacterial loads in sputum (*Sigal et al., 2017*; *Tsao et al., 2000*). Consonantly, high transmission strains induced an increased expression of IL-1β in the lungs of infected mice compared to low transmission ones (*Lovey et al., 2022*). Therefore, our and others' findings indicate that L4 constitutes an example of impeded pro-inflammatory responses not to be the only way for the MTBC members to achieve global success. Lastly, L3 was characterized by a tendency to preferentially remain in a metabolically active state, in contrast with the increased propensity of L2 to enter dormancy, which was previously reported under standard culture conditions in broth (*Reed et al., 2007*). While it was originally thought that this phenotype could be responsible for the increased virulence displayed by L2 Beijing strain in the mouse model, subsequent studies disproved this hypothesis (*Domenech et al., 2017*). In line with this, our results show that even in the context of hypoxic granulomatous responses—which are lacking in the mouse model—there was no correlation between the propensity to switch into a dormant-like state and virulence, as reflected in bacterial burden in our model, across MTBC lineages.

Last but not least, we have identified immune response traits that correlate with the hampered mycobacterial proliferation exhibited by various MTBC strains of independent lineages. T cells are essential for immunity against the MTBC (*Kaufmann, 2002*) hence, it is unsurprising that the level of CD4 and CD8 T cell activation inversely correlated with mycobacterial growth rate. Unfortunately, causality between these two parameters could not be tested experimentally since depleting T cells from PBMCs would imply removing up to 70% of the cells present in the specimen. Increased levels of CXCL9, granzyme B, and TNF were also found to be associated with a better control of MTBC proliferation. Among the identified soluble factors, the prominent protective role of TNF in TB is well known (*Flynn et al., 1995*). CXCL9's paradigmatic function is directing the migration of CXCR3-expressing cells (i.e. T helper 1 CD4 T cells, effector CD8 T cells, and natural killer cells) (*Groom and Luster, 2011*). CXCL9 has attracted attention as potential biomarker for TB diagnosis and treatment monitoring (*Ambreen et al., 2021*; *Uzorka et al., 2022*). A blocking experiment failed to establish a direct relationship between CXCL9 levels and MTBC growth. Warranting further investigations, we hypothesize that chemokines such as CXCL10 and CXCL11 that were not included in our screening panel and that are also signaling through CXCR3 may be acting in concert with CXCL9 to drive this phenomenon. Indeed, some evidence suggests that CXCR3 ligands play a role in the positioning of effector cells within the lung for control of MTB proliferation. Protective vaccination in mice triggered the production of chemokines, including CXCL9, in the lung and associated recruitment of CXCR3+ CD4 T cells (*Khader et al., 2007*). In addition, increased densities of CXCR3+ CD4 T cells were found in the lungs of latently infected NHPs relative to those with active TB (*Shanmugasundaram et al., 2020*). CXCL9 has also been reported as a new marker of trained immunity in mycobacterial growth inhibition assays (*Joosten et al., 2018*). Recently MTBC-exposed individuals, which exhibited the strongest control of mycobacterial proliferation, produced increased levels of CXCL9. Notably, our findings are in line with recent reports highlighting an important role of granzyme B in the control of MTBC infection in the early stages of TB. In NHPs, low burden granulomas were associated with a higher proportion of a T/NK cell cluster expressing granzyme B (*Gideon et al., 2022*). Furthermore, individuals with latent TB infection exhibited increased expression of granzyme B in activated NK cells (*Esaulova et al., 2021*). IFN-γ levels did not exhibit a significant correlation, only a trend, with mycobacterial replication rate in *in vitro* granulomas. Our finding is supported by TB vaccine trials demonstrating that induction of strong IFN-γ responses does not translate into enhanced protection (*Tameris et al., 2013*). However, strains with low replication rates were among those inducing high IFN-γ levels. These results therefore suggest that IFN-γ is required, yet not sufficient to mediate protection.

In summary, our investigation exposed various features that associate with certain MTBC lineages and/or strains, thereby reinforcing the concept that effective treatments and vaccines need to be tailored. We advocate *in vitro* granuloma models as relevant tools to understand the implications of MTBC strain diversity on the interaction with humans of diverse genetic backgrounds and to assess protective traits elicited by vaccines.

## Materials and methods

### Isolation of human PBMCs

Buffy coats from healthy, anonymous blood donors (under informed consent) were purchased from the Interregionale Blutspende SRK AG (Bern, Switzerland). PBMCs were isolated from buffy coats by Ficoll-Paque (GE Healthcare) density gradient and washed twice with RPMI-1640 with L-glutamine (RPMI; Sigma-Aldrich). Aliquots were cryopreserved in RPMI supplemented with 10% DMSO (Sigma-Aldrich) and 40% fetal bovine serum (FBS; Gibco) and stored in liquid nitrogen. Isolated PBMCs were systematically tested for CD4 T cell reactivity against purified protein derivative (PPD; Statens Serum Institute, RT23) as previously reported by our laboratory (*Arbués et al., 2020a*). Prior to use, PBMCs from PPD-reactive donors were thawed, washed twice in RPMI containing 10% FBS (RPMI-FBS) and benzonase (12.5 U/ml; BioVision), and rested in RPMI-FBS overnight at 37°C-5% $CO_2$. Trypan blue dye exclusion method was used to confirm sample viability of above 95%. PBMC concentration was adjusted to $10^7$ cells/ml in RPMI supplemented with 20% human AB serum (PAN-Biotech) (RPMI-HS).

### Culture and preparation of disperse suspensions of MTBC strains

We selected 14 isolates (*Table 1*) from a reference set of clinical strains covering much of the global diversity of the human-adapted MTBC (*Borrell et al., 2019*). This strain collection comprises drug-susceptible strains isolated from patients of diverse origin that can be successfully grown in culture. The strain subset used encompasses three strains per lineage for L1 to L4 and two L5 strains. The laboratory strain H37Rv, which was used for the implementation of the model (*Arbués et al., 2020a*), was included as a reference. We excluded L6 from the present study due to its characteristic significantly slower growth in axenic culture (*Gehre et al., 2013*). L7 to L10 were not included since they have only been described recently and the number of available isolates is very scarce.

MTBC strains were cultured under gentle agitation in Middlebrook 7H9 broth supplemented with 10% ADC (5% bovine albumin fraction V, 2% dextrose and 0.003% catalase), 0.5% glycerol (PanReac AppliChem), and 0.05% Tween-80 (Sigma-Aldrich), additionally enriched with 40 mM sodium pyruvate (Sigma-Aldrich) for L5 strains. Upon reaching mid-exponential phase ($OD_{600}$ between 0.4 and 0.8), bacteria were washed with PBS containing 0.1% Tween-80 (PBST) and resuspended in RPMI-FBS. The resultant mycobacterial suspension was dispersed by water-bath sonication (XUB5, Grant Instruments) for 2 min and then centrifuged at 260×*g* for 5 min. The upper part of the supernatant was recovered and cryopreserved by adding 5% glycerol (final) and stored at –80°C. Concentration of the frozen stocks was quantified by CFU assessment (see below).

### Generation and quantification of 3D *in vitro* granulomas

PBMCs from four healthy blood donors were infected independently with the selected MTBC isolates, and subsequently embedded within an extracellular matrix (ECM) composed of collagen and fibronectin, the main components of the lung ECM, as previously reported (*Arbués et al., 2020b*). Briefly, an ECM solution composed of 950 µl/ml of collagen (PureCol, 3 mg/ml; Advanced BioMatrix), 50 µl/ml of PBS 10×, 4 µl/ml of human fibronectin (1 mg/ml; Sigma-Aldrich), and 10 µl of 1 N NaOH (Sigma-Aldrich) was prepared and incubated at 4°C for 90 min. Rested PBMCs were infected with the various MTBC strains at a multiplicity of infection of 1:200 (bacteria:PBMCs), or left uninfected when required. Thereafter, $1.25×10^6$ PBMCs per well were distributed across 48-well plates and mixed with the preincubated ECM in a 1:1 (v/v) ratio. The ECM was left to set for 45 min at 37°C-5% $CO_2$ before 250 µl of RPMI-HS were added.

Granuloma formation was monitored on day 7 postinfection using a Leica THUNDER Imager Live Cell (Leica). Cellular aggregates present in a representative bright field per donor and condition were enumerated and measured using ImageJ 1.52n (National Institutes of Health, USA). The total number of granulomas, their size, and aspect ratio (i.e. the ratio of the major axis to the minor axis; and so, an aspect ratio of 1 corresponds to a circle, and >1 to an ellipse) were integrated into a aggregate formation score as follows: the area of each individual granuloma was divided by its corresponding aspect ratio, then the average was calculated and divided by the total number of aggregates.

$$Aggregate\,formation\,score = \frac{\overline{X}\left(aggregate\,area/aspect\,ratio\right)}{No.\,aggregates}$$

Thus, the induction of numerous but small aggregates with an elongated shape translates into the lowest aggregate formation scores, while generation of bigger, even if fewer, circular aggregates results in higher values.

## Retrieval of mycobacteria

At the relevant time points, supernatant was removed and the ECM was digested with 125 µl of collagenase (1 mg/ml; Sigma-Aldrich) for 40 min at 37°C-5% $CO_2$. Subsequently, host cells were lysed by adding 125 µl of 0.4% Triton X-100 (Sigma-Aldrich) and incubating for 20 min at room temperature. Released bacilli were subsequently used for bacterial load quantification by CFU assessment and dual Au/NR staining.

## CFU assessment

Briefly, 10-fold serial dilutions of the mycobacterial suspensions were prepared in triplicate in PBST and plated on 7H11 agar (BBL) plates supplemented with 0.5% glycerol and 10% OADC (0.05% oleic acid in ADC), additionally enriched with 40 mM sodium pyruvate for L5 strains. Plates were incubated at 37°C-5% $CO_2$ for 3–4 weeks.

## Dual Au/NR staining

Bacilli retrieved on day 7 post-infection were inactivated with 1× CellFIX (BD) for 20 min at room temperature and afterward stored at 4°C until processing. Fixed samples were centrifuged at 10,000×$g$ for 5 min to remove the fixative prior being resuspended in water. Subsequently, they were spotted on glass slides, air-dried, and heat-fixed at 70°C. Fluorescent acid-fast staining using TB Fluorescent Stain Kit M (BD) was performed in combination with neutral-lipid staining dye NR (Sigma) (*Arbués et al., 2020a*). Briefly, each sample was stained with Au for 20 min, decolorized for 30 s, covered with NR (10 µg/ml) for 15 min and counterstained with potassium permanganate for 2 min, including gentle washes with distilled water between each step. Air-dried, stained slides were mounted using Vectashield mounting medium (Vectorlabs) and examined using a Leica DM5000 B fluorescence microscope (Leica). For quantification purposes, at least 200 bacteria per sample were counted manually.

## Macrophage cell death quantification

On day 6 post-infection, cells were released from the ECM by collagenase digestion as described above. Subsequently, they were pelleted at 400×$g$ for 5 min and extracellularly stained as follows. Cells were incubated for 15 min at room temperature in 100 µl of Annexin binding buffer (ABB, BioLegend) containing APC-labeled Annexin V, 7-AAD, and fluorochrome-labeled antibodies (BioLegend) against human CD3 (Pacific Blue, clone UCHT1, Cat# 300431, RRID:AB_1595437), CD19 (BV510, clone HIB19, Cat# 302242, RRID:AB_2561668), and CD11b (PE-Cy7, clone ICRF44, Cat# 301322, RRID:AB_830644). Samples were washed once with ABB and fixed in 1× CellFIX (BD) for 20 min at room temperature. Samples were acquired on a MACSQuant Analyzer 10 (Miltenyi Biotec) and processed using FlowJo 10.6.1 (BD).

## Assessment of lymphocyte activation and proliferation

Prior to infection for the generation of 3D *in vitro* granulomas, rested PBMCs were labeled with CFSE. Briefly, PBMCs were resuspended in PBS containing 1 µM CFSE (BioLegend) at 5×10$^6$ cells/ml and incubated at 37°C-5% $CO_2$ for 10 min. After quenching the staining by adding 2.5 volumes of PBS containing 20% FBS, cells were pelleted and resuspended at 10$^7$ cells/ml in RPMI-HS.

On day 6 post-infection, cells were released from the ECM by collagenase digestion as described above. Subsequently, they were pelleted at 400×$g$ for 5 min and extracellularly stained following a standard protocol. Cells were incubated for 20 min at room temperature in 100 µl of FACS buffer (0.1% FBS in PBS) containing fluorochrome-labeled antibodies (BioLegend) against human CD3 (Cat# 300431, RRID:AB_1595437, clone UCHT1, Pacific Blue), CD4 (Cat# 300512, RRID:AB_314080, clone RPA-T4, PE-Cy7), CD8 (Cat# 300934, RRID:AB_2814115, clone HIT8a, BV510), CD19 (Cat# 302242, RRID:AB_2561668, clone HIB19, BV510), CD25 (Cat# 302606, RRID:AB_314276, clone BC96, PE), CD69 (Cat# 310928, RRID:AB_10679124, clone FN50, PerCP), CD38 (Cat# 303510, RRID:AB_314362, clone HIT2, APC), and HLA-DR (Cat# 307618, RRID:AB_493586, clone L243, APC-Cy7). Samples were washed once with FACS buffer and fixed in fixation buffer (BioLegend) for 20 min at room

temperature. Samples were acquired on a MACSQuant Analyzer 10 (Miltenyi Biotec) and processed using FlowJo 10.6.1 (BD).

## Quantification of soluble factors

Supernatants collected on day 1 and 8 post-infection, and stored at –80°C in the meantime, were filter-sterilized no more than 24 h prior to analysis by multiplex bead-based immunoassay. A ProcartaPlex custom panel (including CCL2/MCP-1, CCL3/MIP-1α, CXCL9/MIG, TNF, IL-1β, IL-2, IL-6, IL-10, IL-12p70, IL-13, IFN-α, IFN-γ, GM-CSF, granzyme B, and MMP-1) (Invitrogen) and the Bio-Plex Pro human Th17 cytokine panel (Bio-Rad) (including IL-17A, IL-17F, IL-21, IL-22, and IL-23) were used according to the manufacturer's recommendations and acquired on a Luminex Bio-Plex 200 platform and Bio-Plex Manager 6.0 software (Bio-Rad). Culture medium was measured in parallel to the samples to assess any potential residual presence of the target biomolecules in the human serum. Results were analyzed using R package nCal (*Fong et al., 2013*).

## Blocking of IL-1β and CXCL9 early responses

In order to evaluate the impact of blocking IL-1β response on granuloma formation, PBMCs were treated on infection day with 200 ng/ml of anti-IL-1β biologic gevokizumab (Ichorbio, Cat# ICH5133, RRID:AB_307577) or a human IgG2 isotype control (BioLegend, Cat# 403601, RRID:AB_3097033, clone QA16A13). On day 4 post-infection, supernatant was removed and replenished with fresh, antibody-containing RPMI-HS. Granuloma formation was monitored on day 7 post-infection and aggregate formation score calculated as described in the Generation and quantification of 3D *in vitro* granulomas section.

Similarly, to assess the effect of blocking CXCL9 on bacterial proliferation, PBMCs were treated on infection day with 200 ng/ml of a neutralizing antibody (BioLegend, Cat# 943503, RRID:AB_2894528, clone A16023A) or a mouse IgG1 isotype control (BioLegend, Cat# 401407, RRID:AB_11148722, clone MG1-45). On day 4 post-infection, supernatant was removed and replenished with fresh, antibody-containing RPMI-HS. Bacterial load was quantified on day 8 post-infection as described in the Retrieval of mycobacteria and CFU assessment sections.

## Quantification and statistical analysis

GraphPad Prism 8.2.1 and R.4.3.1 and RStudio 2023.06.1-524 were used to produce quantitative graphical representations of the generated data and to perform statistical analyses. The nature of the tests and definition of center and significance levels is specified within the respective figure legend.

## Acknowledgements

This work was supported by the Swiss National Science Foundation (SNSF197838).

---

## Additional information

### Funding

| Funder | Grant reference number | Author |
|---|---|---|
| Schweizerischer Nationalfonds zur Förderung der Wissenschaftlichen Forschung | SNSF197838 | Damien Portevin |

The funders had no role in study design, data collection and interpretation, or the decision to submit the work for publication.

### Author contributions

Ainhoa Arbués, Conceptualization, Formal analysis, Investigation, Visualization, Methodology, Writing – original draft, Writing – review and editing; Sarah Schmidiger, Investigation, Visualization, Methodology, Writing – original draft, Writing – review and editing; Miriam Reinhard, Investigation; Sonia

Borrell, Sebastien Gagneux, Writing – review and editing; Damien Portevin, Conceptualization, Formal analysis, Supervision, Funding acquisition, Visualization, Methodology, Writing – original draft, Writing – review and editing

#### Author ORCIDs
Ainhoa Arbués https://orcid.org/0000-0002-3377-4171
Sarah Schmidiger https://orcid.org/0009-0008-1791-4104
Damien Portevin https://orcid.org/0000-0003-2949-9557

Reviewer #2 (Public review): https://doi.org/10.7554/eLife.99062.4.sa1
Reviewer #3 (Public review): https://doi.org/10.7554/eLife.99062.4.sa2
Author response https://doi.org/10.7554/eLife.99062.4.sa3

## Additional files

#### Supplementary files
MDAR checklist
Source data 1. Raw data generated during this study.

#### Data availability
The data generated during this study are available as *Source data 1*.

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
