## [Editor Report · eLife Assessment]

This study describes the impact of mycobacterial genetic diversity on host-infection phenotypes by assessing the effect of different *M. tuberculosis* lineages on granulomatous inflammation using a 3D in vitro granuloma model. Despite being descriptive and showing mostly correlative relationships, the findings are **useful** and provide some **solid** support regarding the functional impact of M. tuberculosis's natural diversity on host-pathogen interactions. The study will interest researchers working on mycobacteria and motivate future studies to understand how genetic diversity influences virulence and immunity outcomes.

---

## [Referee Report · Reviewer #2 (Public review)]

Summary:

This manuscript reports a comparison of microbial traits and host response traits in a laboratory model of infected granuloma using Mtb strains from different lineages. The authors report increased bacillary growth and granuloma formation, inversely associated with T cell activation that is characterized by CXCL9, granzyme B and TNF expression. They therefore infer that these T cell responses are likely to be host-protective and that the greater virulence of modern Mtb lineages may be driven by their ability to avoid triggering these responses.

Strengths:

The comparison of multiple Mtb lineages in a granuloma model that enables evaluation of the potential role of multiple host cells in Mtb control, offers a valuable experimental approach to study the biological mechanisms that underpin differential virulence of Mtb lineages that has been previously reported in clinical and epidemiological studies.

Weaknesses:

The study is rather limited to descriptive observations, and lacks experiments to test causal relationships between host and pathogen traits. Some of the presentation of the data are difficult to interpret, and some conclusions are not adequately supported by the data.

Comments on revisions:

The authors have addressed my previous comments with appropriate revisions and explanations.

---

## [Referee Report · Reviewer #3 (Public review)]

Arbués and colleagues describe the impact of mycobacterial genetic diversity on host-infection phenotypes. The authors evaluate Mtb infection and contextualize host-responses, bacterial growth and metabolic transitioning in vitro using their previously established model of blood-derived, primary-human-cells cultured within a collagen/fibronectin matrix. They seek to demonstrate the effectiveness of the model in determining mycobacterial strain specific granuloma-dependent host-pathogen interactions.

Understanding the way mycobacterial genetic diversity impacts granuloma biology in tuberculosis is an important goal. One of this works strengths is the use of primary human cells and two constituents of pulmonary extracellular matrix to model Mtb infection. The authors and others have previously shown that Mtb infected PBMC aggregates share important characteristics with early pulmonary TB granulomas. Use of multiple genetically distinct strains of Mtb defines this work and further bolsters it potential impact. However, the study is not comprehensive as lineages 6 and 7 are not tested. Experiments are primarily descriptive, and the methodologies are conventional. Correlative relationships are the manuscripts focus and effect sizes are generally small.

The main aim of this work is to extend an in vitro granuloma model to the study of a large collection of well characterized, genetically diverse representatives of the *Mycobacterium tuberculosis* complex (MTBC). I believe that they accomplish that aim. The work does investigate MTBC infection of aggregated PBMCs using three strains each of Mtb lineages 1-5 and H37Rv, which is not a trivial undertaking. The experimental aims are to show that MTBC genetic diversity impacts growth and dormancy of granuloma bound bacteria and, the host responses of granulomatous aggregation as well as macrophage apoptosis, lymphocyte activation and soluble mediator release within granulomas. The methodologies employed are sufficient to test most of these aims. The authors conclusions regarding their results are mostly supported by the data. The conclusion that lineage impacts growth within granulomas is likely true and the data as presented reflect such a relationship. Their conclusions regarding lineage's impact on dormancy are partially supported, as their findings demonstrate that assays for dormancy identify strain-specific metabolic changes in the bacteria consistent with a dormancy-like state but also identify replicating bacteria as being dormant. The data strongly supports the impact of mycobacterial genetic diversity on a spectrum of granulomatous responses in their model system. Those findings are a highlight of the publication. The data further supports the idea that strain diversity impacts macrophage apoptosis but a relationship of apoptosis to the granulomatous response is not effectively evaluated. The association of lymphocyte activation with reduced mycobacterial growth as an aspect of granulomas is well documented in the literature and a negative correlation between T cell activation and growth is supported by the authors results. Their data also support the conclusion that soluble mediator production by PBMCs is different based on the infecting strain of mycobacteria and that IL1b modulates aggregate phenotypes in their model.

The authors contribute some valuable insights, particularly in Figure 3. Their model is higher echelon relative to others in the field, but I don't believe that it possesses all the components necessary to replicate formation of mycobacterial granulomas in vivo. That being said, their identification of donor-dependent aggregation phenotypes by mycobacterial strain has the potential to enable future investigations of human and mycobacterial genetic components that are involved in the formation of TB granulomas.

---

## [Author Response]

The following is the authors’ response to the previous reviews.

**Reviewer #2:**
The authors indicated that they had added coefficients of variation for within-lineage heterogeneity (line 93), but I can't seem to find this.

The coefficients of variation were indeed included as suggested, and can be found in lines 94-96 of the current revised version of the manuscript. The sentence states: “Nevertheless, substantial intra-lineage heterogeneity could be observed, particularly within L1 and L2 (coefficients of variation 84.4% [L1] and 66.0% [L2] vs. 32.6% [L3], 34.6% [L4] and 31.9% [L5]).”

They were unable to address my question on the impact of T-cell depletion from PBMC on bacterial growth? Their discussion should include that this experimental limitation means that they are unable to test cause and effect for the relationship between T cell proliferation and bacterial growth.

As recommended, this experimental limitation is now included in the discussion in lines 344-346.

**Reviewer #3:**
EM:Based on the authors lack of resources, I don't believe that electron microscopy experiments should be required for this publication. However, it should be noted that EM is performed on fixed samples such that implementation of those protocols as it relates to bio-safety is no more demanding than the preparation of samples for other common assays performed outside of the BSL3.

We appreciate your understanding regarding our lack of resources to carry out the EM experiments, although we recognize the possibility of them being performed on BSL3 samples.

Granuloma score:From the author comments and the manuscript's text, it appears that the "granuloma score" is an attempt at quantitation of PBMC organization. Where every component of the metric [(mean area / mean aspect ratio) / mean n] is a visual facet of the relative integration of PBMCs into a more organized aggregate. The area and number (n) of aggregates both address regional coalescence of the total number of PBMCs added into the matrix. Whereas the aspect ratio component is an indicator of uniformity of the PBMCs that have been assigned to an individual aggregate. Perhaps another roundness estimation would have been a more precise, but aspect ratio seems fine for their assay. Considering these factors and the author's contention that the aggregates making up (n) are granulomas, the name "granuloma score" is inaccurate and a more appropriate title would be "aggregate organization score" or "aggregate organization index".

Thank you for the suggested alternative terminology, the term “granuloma score” has been substituted with “aggregate formation score” throughout the manuscript.

Dormancy:In the manuscript, the authors should explicitly reference the validation studies which demonstrate induction of the DosR regulon in the model, lest their previously generated and conducted studies go unappreciated by a broader audience. In the title of that previous work (PMID: 32069329) this group used the designation "dormant-like" to describe the state observed in bacteria within their in vitro granuloma model system, as they also do in LINE 124. This term or a variation of it should be exchanged for dormant/dormancy throughout the manuscript when referring to observations in the model bacteria. It is a more precise description. Further, "dormant-like" allows the latitude to refer to actively growing bacteria in the context of dormancy without running the risk of putting forth confusing or potentially erroneous assertions.

As recommended, the suffix “-like” has been added to the designation “dormant” when referring to the bacterial phenotype induced in the model. In addition, de induction of the DosR regulon in the model is now mentioned in line 116 and the reference to Kapoor’s work that originally demonstrated it by qPCR included.

PBMC aggregation:I would like to make the authors aware that in well vetted models, cell aggregation as a function of infection does not typically occur in PBMCs on tissue culture plates until day 6 post infection (PMID: 25691598, Fig 2). Further, this group's own published protocol for the model under consideration in this manuscript (PMID: 33659472, Fig1) explicitly states that "Formation of granuloma like structures can be observed after 7-8 days", the implication being that prior to 7 days granuloma like structures cannot be observed reliably. Regardless, it seems evident that the authors will not be conducting additional experiments for this publication, which I find acceptable. However a proper negative control would certainly strengthen evidence for the association of strain specific bacterial and host responses with the granulomatous response in this model.

We had interpreted the reviewer’s previous comment regarding PBMC aggregation as referring to a different experimental model rather than a matter of timing. Since many other studies have previously assessed the impact of strain/lineage variability in macrophage responses, in this work we decided to focus on later time points and we did include uninfected as a negative control. Nonetheless, we agree it would be indeed very interesting to additionally evaluate monocyte/macrophage early responses and we will take it into account for future studies.

Use of antiquated terminology:I can appreciate the desire to establish continuity between publications by using the same abbreviation for TNF but it will come at a cost. Using outdated terms in general makes people more dismissive of the work. Perhaps something to consider.

Since this seems an important issue to the reviewer, we have replaced the term TNF-α with TNF throughout the manuscript.